# Association between Albumin Alterations and Renal Function in Patients with Type 2 Diabetes Mellitus

**DOI:** 10.3390/ijms25063168

**Published:** 2024-03-09

**Authors:** Marta Nugnes, Maurizio Baldassarre, Danilo Ribichini, Daniele Tedesco, Irene Capelli, Daniele Vetrano, Francesca Marchignoli, Lucia Brodosi, Enrico Pompili, Maria Letizia Petroni, Gaetano La Manna, Giulio Marchesini, Marina Naldi, Manuela Bartolini

**Affiliations:** 1Department of Pharmacy and Biotechnology, Alma Mater Studiorum University of Bologna, 40126 Bologna, Italy; marta.nugnes2@unibo.it (M.N.); daniele.tedesco@isof.cnr.it (D.T.); manuela.bartolini3@unibo.it (M.B.); 2Unit of Semeiotics, Liver and Alcohol-Related Diseases, IRCCS Azienda Ospedaliero-Universitaria di Bologna, 40138 Bologna, Italy; maurizio.baldassarre@unibo.it (M.B.); enrico.pompili3@unibo.it (E.P.); 3Endocrinology and Diabetes Prevention and Care Department, IRCCS Azienda Ospedaliero-Universitaria di Bologna, 40138 Bologna, Italy; danilo.ribichini@aosp.bo.it; 4Institute for Organic Synthesis and Photoreactivity, National Research Council, 40129 Bologna, Italy; 5Department of Medical and Surgical Sciences, Alma Mater Studiorum University of Bologna, 40126 Bologna, Italy; irene.capelli4@unibo.it (I.C.); daniele.vetrano@studio.unibo.it (D.V.); gaetano.lamanna@unibo.it (G.L.M.); 6Nephrology, Dialysis and Renal Transplant Unit, IRCCS Azienda Ospedaliero-Universitaria di Bologna, 40138 Bologna, Italy; 7Unit of Clinical Nutrition, IRCCS Azienda Ospedaliero-Universitaria di Bologna, 40138 Bologna, Italy; francesca.marchignoli@gmail.com (F.M.); lucia.brodosi2@unibo.it (L.B.); marialetizia.petroni@unibo.it (M.L.P.); giulio.marchesini@unibo.it (G.M.); 8Centre for Applied Biomedical Research (CRBA), Alma Mater Studiorum University of Bologna, 40126 Bologna, Italy

**Keywords:** diabetic kidney disease, effective albumin, reduced albumin, structural alterations, oxidative damage, high-resolution mass spectrometry

## Abstract

Diabetic kidney disease (DKD) is a major cause of morbidity and mortality in individuals with type 2 diabetes mellitus (T2DM). The aim of this study was to investigate whether albumin structural alterations correlate with DKD severity and evaluate whether native and reduced albumin concentrations could complement the diagnosis of DKD. To this end, one hundred and seventeen T2DM patients without (*n* = 42) and with (*n* = 75) DKD (DKD I-III upon KDIGO classification) were evaluated; the total albumin concentration (tHA) was quantified by a bromocresol green assay, while structural alterations were profiled via liquid chromatography–high-resolution mass spectrometry (LC-HRMS). The concentrations of native albumin (eHA, effective albumin) and reduced albumin (rHA) were subsequently assessed. The HRMS analyses revealed a reduced relative amount of native albumin in DKD patients along with an increased abundance of altered forms, especially those bearing oxidative modifications. Accordingly, both eHA and rHA values varied during the stages of progressive renal failure, and these alterations were dose-dependently correlated with renal dysfunction. A ROC curve analysis revealed a significantly greater sensitivity and specificity of eHA and rHA than of tHA for diagnosing DKD. Importantly, according to the multivariate logistic regression analysis, the eHA was identified as an independent predictor of DKD.

## 1. Introduction

Diabetic kidney disease (DKD) is one of the most common chronic complications and a major cause of morbidity and mortality in patients with diabetes mellitus (type 1, T1DM; type 2, T2DM) [1,2,3]. Among 400 million T2DM patients worldwide, 50% show evidence of chronic kidney disease (CKD), which is mainly related to DKD. According to regional studies, the incidence of DKD in the diabetic population ranges from 30% to 80% or more [4]. Moreover, DKD is considered the leading cause of end-stage kidney disease (ESKD), and DKD patients account for 25–45% of all patients enrolled in ESKD programs [2]. This is particularly worrying considering that the prevalence of diabetes mellitus (especially T2DM) has risen dramatically worldwide; in 2021, 11% of the global population had diabetes, and this prevalence is expected to reach 12% by 2045 [4]. The early identification of DKD is a primary unmet clinical need, not only for predicting and preventing disease progression, but also for improving patient survival and reducing associated morbidities. As for KDIGO guidelines [5], the diagnosis of kidney damage is based upon the observation of decreased renal function with an estimated glomerular filtration rate (eGFR) < 60 mL/min/1.73 m^2^ or the presence of markers of renal impairment, such as albuminuria, hematuria, or abnormalities detected by laboratory or imaging tests, present for at least for at least 3 months and with health consequences [6].

These parameters are frequently associated with elevated blood pressure and cardiovascular complications, which are major causes of morbidity and mortality. Renal biopsy is not routinely suggested by clinical guidelines to assess DKD [7]; however, renal histological assessment can be valuable for identifying kidney pathologies other than diabetic disease, including nondiabetic renal disease (NDRD) [2,8], and for studying and classifying diabetic lesions in patients to stratify prognosis and guide treatment [9].

In 2012, a cross-sectional study carried out on a cohort of 15,773 T2DM patients suggested that patients with significant albuminuria predominantly experience microvascular complications—the kidney being the main target of microvascular damage in diabetes—while cardiovascular complications were principally associated with reduced eGFR alone [10].

The prototype of DKD is characterized by an early stage of glomerular hyperfiltration, followed subsequently by the onset of albuminuria and later by the progressive decline in the eGFR. However, additional phenotypes have now been identified, some of which are characterized by the presence of microalbuminuria alone or the absence of urinary protein excretion. These phenotypes are often characterized by a rapid decline in renal function [11]. In both cases, it is clear that albuminuria, per se, is not sensitive enough as a biomarker in the early phase of DKD, whereas the eGFR is proven to risk-stratify DKD patients only when it is <60 mL/min/1.73 m^2^, i.e., when almost half of the kidney function is lost [12,13]. Furthermore, the prediction of the decline of kidney function and recognition of risk factors, provided by current parameters, remain imprecise.

In this scenario, the use of more sensitive biomarkers or a combination of multiple biomarkers reflecting different aspects of the pathophysiology of kidney impairment may improve patient stratification, help in the early diagnosis of DKD, and prompt appropriate therapeutic intervention. Indeed, identifying these biomarkers has been the subject of intense investigations. In recent decades, several biomarkers, mainly proteins, have been proposed [12,14,15,16]; however, most of them lack rigorous external validation in adequately powered studies with renal endpoints [14].

During its relatively long plasmatic half-life (16–19 days), albumin undergoes several structural modifications including oxidation, glycation, and truncation [17]. Although these modifications are also encountered in healthy patients, their extent is significantly increased in patients with chronic diseases characterized by increased proinflammatory and pro-oxidant circulatory microenvironments (Figure 1) [18,19,20]. A recent study in patients with liver cirrhosis showed that a decrease in the total serum albumin concentration (tHA), a common feature of this disease, is accompanied by significant structural alterations, mainly oxidation at the only free cysteine residue (Cys34) and truncations [21]. Due to the high plasma concentration of albumin, the reduced form of Cys34 represents the main plasma reservoir of free thiol groups, which are endowed with scavenging capacity [22]. As a result of these structural alterations, the native form of the protein (nHA) was decreased to a greater extent than was tHA. From this observation, the concept of the effective albumin concentration (eHA), namely, the concentration of albumin in its native form, was introduced [21,23]. In the context of decompensated cirrhosis, eHA has been shown to be endowed with greater diagnostic and prognostic power than tHA. Hypoalbuminemia is also considered a risk factor for end-stage CKD because it increases morbidity and mortality in renal failure patients [24,25,26,27]. Moreover, previous studies have shown that the albumin structure is partially altered in T2DM patients with renal impairment and proposed the oxidized form of the protein as a marker for disease progression [28,29].

In the present work, a liquid chromatography–high-resolution mass spectrometry (LC-HRMS) approach was used in combination with a standard colorimetric assay to assess eHA and the concentration of all albumin forms reduced at the level of Cys34 (rHA) in a cohort of T2DM patients with and without DKD. The aim of this study was to assess whether eHA and/or rHA parameters can complement the diagnosis of renal impairment.

## 2. Results

### 2.1. Subject Population

One hundred and seventeen patients with a diagnosis of T2DM for at least 1 year were enrolled between 2018 and 2021. The patients were classified into control or DKD groups according to their albuminuria and eGFR values following the current Clinical Practice Guideline for Diabetes Management in Chronic Kidney Disease (KDIGO) [5,30]. Patients with an eGFR greater than 60 mL/min/1.73 m^2^ were tested as controls, while patients with an eGFR < 60 mL/min/1.73 m^2^ and a UACR ≥ 30 mg/g were considered to have DKD. The anthropometric and clinical characteristics of the enrolled patients with or without DKD are reported in Table 1. Briefly, patients with DKD were older and more likely to be on blood-pressure-lowering and diuretic therapies. In addition, DKD patients were characterized by lower cholesterol, LDL, and HDL levels and higher triglyceride levels. As a result, the triglyceride-to-HDL ratio was significantly greater in patients with DKD. As expected, DKD patients were also characterized by more impaired renal function, as indicated by higher creatinine levels, lower eGFR value, and higher UACR (Table 1).

DKD patients were further classified according to the degree of renal damage following the CGA scheme proposed by KDIGO, where CGA stands for cause, GFR category (G1–G5), and albuminuria category (A1–A3), into moderately increased risk (DKD-I), high risk (DKD-II), and very high risk (DKD-III) of poor prognosis (see details in Figure 2).

The anthropometric and clinical characteristics of the DKD patients grouped according to risk category are listed in Appendix A. The incidence of DKD I-III was similar with respect to age, sex, BMI, and most pharmacologic treatments, except for glucose-lowering drugs. The three groups also had significantly different lipid profiles (Appendix A). Within the DKD-III group, eight patients (25%) were receiving dialysis at the time of enrollment (patients with an eGFR < 15 were included in G5; Table 1).

### 2.2. Evaluation of Albumin Structure

The high-resolution MS-based analytical approach allowed the assessment of the relative amounts of the (i) native form of albumin (nHA), (ii) albumin (HA) forms with Cys34 in the reduced form (mercaptoalbumin; HMA), (iii) HA forms with Cys34 in the cysteinylated form (non-mercaptoalbumin 1; HNA1), (iv) HA forms with irreversibly oxidized Cys34 (non-mercaptoalbumin 2; HNA2), (v) glycated HA forms, (vi) HA forms with truncations at the C- and N-terminals, and (vii) HA carrying combinations of these alterations (Figure 2, Table 2). The results highlighted a significantly greater number of structural alterations in the circulating HA from DKD patients and showed that these alterations primarily involved the redox state of Cys34 (Figure 3). Specifically, in patients with kidney damage, the percentage of HNA1 increased by approximately 9% (from 16.4 (13.7–19) % to 25.7 (19.8–30.9) %; *p* < 0.0001). Consistently, a significant reduction in the native form, i.e., nHA (from 59.0 (57.1–60.6) % to 52.5 (48.3–56.7) %; *p* < 0.0001), and HMA (from 75.7 (73.2–78.4) % to 67.1 (62.5–72.7) %; *p* < 0.0001) was observed. A slight, but significant, decrease in HNA2 was also observed (Table 2). Apart from the redox state of Cys34, no significant difference in the abundance of the truncated forms was observed. Conversely, a slight increase (from 10.8 (9.2–12.2) % to 12.5 (10.5–14.7) %, *p* = 0.0028) in the relative abundance of glycated albumin was detected (Table 2). Notably, carbamylated HA was previously annotated as an altered form of HA in DKD patients [31]; however, in our samples, no significant amount of carbamylated HA was detected. Considering that both the HA concentration and structural integrity are altered in the presence of renal damage and that most alterations involve the redox state of Cys34, we focused our attention on the serum concentration of HMA, i.e., the concentration of all HA forms carrying reduced Cys34. Notably, the observed decrease in HMA levels (−9%) in DKD patients paralleled the increase in HNA1 levels (+9%) (Table 2), as these two changes are related to the redox state of Cys34. Therefore, further investigations were conducted considering only rHA, as similar results of the opposite sign could be obtained with respect to the concentration of HNA1.

### 2.3. Total, Effective, and Reduced Albumin Concentrations

The total albumin concentration (tHA) in the DKD patients [4.1 (3.9–4.4)] g/dL was slightly lower than that in the control patients [4.3 (4.1–4.5)] g/dL (*p* = 0.001) (Table 3). Furthermore, the serum concentrations of both eHA and rHA, representing the serum concentrations of nHA and HMA, respectively, were significantly lower in T2DM patients with kidney damage. These parameters were able to discriminate the patients from the control group better than tHA (*p* < 0.001, Table 3).

ROC curves were plotted to assess the sensitivity and specificity of tHA, eHA, and rHA for diagnosing renal impairment. The area under the curve (AUC) of both eHA and rHA had greater diagnostic power for renal impairment (AUC-eHA: 0.831, 95% CI, 0.760–0.903; AUC-rHA: 0.823, 95% CI, 0.749–0.898) than tHA (AUC-tHA: 0.676, 95% CI, 0.579–0.772; *p* < 0.001) (Figure 4). The cutoff values associated with the highest sensitivity and specificity for eHA and nHA were 2.3 g/dL and 3.2 g/dL, respectively. Interestingly, 87% of patients with DKD had nHA values less than 3.2 g/dL, while this percentage increased to 96% in patients with eHA values less than 2.3 g/dL. Moreover, according to the multivariate logistic regression analysis, only eHA remained an independent predictor of renal impairment (OR 0.58; 95% CI, 0.46–0.73).

### 2.4. Correlation of tHA, rHA, and eHA with Biochemical Parameters

The correlations between tHA, rHA, and eHA and the currently used biochemical parameters of DKD severity, i.e., creatinine, albuminuria, and eGFR, were also evaluated. At baseline, all albumin-related parameters, namely, tHA, rHA, and eHA, were negatively correlated with creatinine and albuminuria levels, and positively with eGFR values (Table 4). After a year, significant correlations with biochemical parameters were maintained for rHA and eHA, while tHA was only associated with the albuminuria level (Table 4).

### 2.5. Albumin Levels and Severity of Diabetic Kidney Disease

The associations of tHA, rHA, and eHA with the progression of DKD were also evaluated. Eight patients receiving dialysis therapy at enrollment, who were previously included in group III, were considered a separate group for the aim of this analysis.

As shown in Figure 5, the tHA was different between patients with minimally impaired renal function (DKD-I) and those with DKD-II, DKD-III, or dialysis (*p* < 0.05), but was not able to distinguish more severe renal disease stages; rHA was able to distinguish the dialysis group from DKD-I and DKD-II (*p* < 0.05) and DKD-III from DKD-I (*p* < 0.05); finally, eHA was able to discriminate DKD-III and dialysis from DKD-I. In summary, rHA and eHA provided additional information for classifying patients according to the stages of their renal disease.

## 3. Discussion

This study showed that the HA structure is impaired in DKD patients, and the incidence of oxidative damage progressively increases with the increasing severity of renal damage, likely resulting from the pro-oxidant environment associated with diabetes. This alteration occurs independently of metabolic control, considering that no significant differences in HbA1c were demonstrated in the present setting; although it was achieved with different pharmacologic treatments, the level of glycated albumin was moderately increased in DKD patients.

CKD can itself be considered a chronic inflammatory disease independently of the presence of DM. In fact, persistent, low-grade inflammation is now widely acknowledged as a pivotal factor in the pathophysiology of renal disease. This inflammatory state assumes a distinctive role, not only contributing to the progression of DKD, but also playing a crucial role in the increased risk of cardiovascular events and all-cause mortality associated with this condition. Furthermore, this chronic inflammatory milieu is implicated in the genesis of protein–energy wasting, further exacerbating the complexities of DKD management. A multitude of factors contribute to the chronic inflammatory state in DKD. These include the increased production and decreased clearance of proinflammatory cytokines, oxidative stress, acidosis, chronic and recurrent infections, the altered metabolism of adipose tissue, and intestinal dysbiosis. The level of inflammation is directly correlated with the eGFR in CKD patients and intensifies in dialysis patients [32,33].

HA is an acute-phase reactant that undergoes several structural modifications during its circulatory life [17]. Although these modifications are also found in healthy patients, their extent is significantly greater in patients with chronic diseases characterized by an increased proinflammatory and pro-oxidant circulatory microenvironment, as is the case of DKD and T2DM-induced KD [18,19,20,34,35,36]. Indeed, a reduction in tHA is considered a parameter associated with long-term survival in several clinical settings and is considered a strong biomarker of poor outcomes in several diseases [37]. Furthermore, HA plasma levels (tHA) have shown consolidated prognostic power for liver diseases and malabsorption syndromes.

Given the clinical relevance of alterations in both HA plasma levels and structure in acute or chronic pathological conditions [37], we focused our attention on evaluating whether DKD severity in T2DM patients is associated with alterations in HA structure by exploiting an MS-based analytical approach that allowed the fine characterization of HA microheterogeneity (Figure 2, Table 2). Consistently with the findings of previous investigations, the present study showed that DKD is accompanied by a greater incidence of altered forms of the protein and that most changes involve the redox state of Cys34 [28,38,39] (i.e., HNA1 and HNA2; Table 2). Cys34 is a key residue of HA since it represents the major plasma reservoir of free thiol groups and acts as a scavenger of reactive oxygen species (ROS), thus contributing to a large part of the plasma antioxidant capacity [22]. Indeed, the significant increase in the oxidized forms of circulating HA, i.e., HNA1, was paralleled by a significant decrease in nHA and HMA in DKD patients, which is in line with increased oxidative stress [34] and may imply a decreased “buffering capacity” toward further ROS-related damage.

Along with the increase in HNA1, a slight, but significant, decrease in HNA2 was also observed. Similar data were previously reported by Baldassarre et al., who showed that the sulfynylated form of albumin was slightly lower in hospitalized cirrhotic patients than in liver disease outpatients [21].

HA structural impairment was accompanied by a significant decrease in tHA levels, in agreement with previous evidence showing that, in disease states accompanied by increased inflammatory processes, as is the case of DKD, albumin levels decrease as a consequence of reduced hepatic synthesis, increased catabolism, and vascular permeability [25].

Due to the key physiological role of reduced HA as an antioxidant agent, along with rHA (the serum concentration of all HA forms reduced at the level of Cys34), eHA, i.e., the serum concentration of native HA, was also evaluated. This evaluation is supported by the promising results previously achieved in the field of decompensated cirrhosis [21].

A comparison of the tHA, rHA, and eHA values showed that both rHA and eHA were significantly decreased in T2DM patients with DKD. More importantly, both rHA and eHA were able to distinguish the stage of renal damage better than tHA. This observation suggested the importance of considering not only the quantity of circulating protein, but also its structural integrity, and prompted us to investigate the diagnostic capacity of these parameters.

The promising diagnostic power of rHA and eHA was confirmed by the analysis of the ROC curves; indeed, the sensitivity and specificity of eHA and rHA were significantly greater than those of tHA for the diagnosis of renal impairment. Finally, a multivariable logistic regression analysis showed that eHA, but not rHA, was an independent predictor of renal impairment.

These results are consistent with those reported by Maruyama’s group, who showed that HNA1, which indirectly describes the antioxidant capacity of albumin, is the parameter that best correlates with the diagnosis of renal damage [40]. Moreover, the fact that eHA, which reflects the concentration of native and fully functional albumin, is the only independent predictor of renal impairment suggests that functions other than the antioxidant capacity of Cys34 (such as binding and detoxification) may be impaired as the disease progresses.

The ability of HA to snapshot the clinical condition of DKD was further confirmed by the significant correlations between tHA, rHA, and eHA and the biochemical parameters commonly used in clinical settings, i.e., creatinine, eGFR, and albuminuria. This means that both the structural integrity and the amount of albumin are affected by the severity of kidney damage. Interestingly, a similar association was observed when the same parameters were assessed at one year of follow-up, albeit in a limited number of patients, suggesting that rHA and eHA may be associated with disease progression.

In terms of clinical impact, a better understanding of the overall status of T2DM patients with different stages of renal damage might also help clinicians in the decision-making process. Hence, the discriminating power of tHA, rHA, and eHA was evaluated. This comparison confirmed that both rHA and eHA levels significantly varied during the stages of progressive renal failure, suggesting that the initial stage of the disease is characterized by a decrease in the serum albumin concentration, while oxidative damage prevails and impacts the oxidative status of Cys34 as renal damage progresses.

In this study, tHA was the only biomarker that significantly decreased in the early stages of the disease, while rHA and eHA levels decreased significantly with the progression of renal damage. Interestingly, only rHA underwent a further significant reduction in the terminal stage of the disease. Given this perspective, the diagnostic power of tHA in the early stages of CKD and DKD is intriguing, especially when used in conjunction with traditional markers of renal damage such as eGFR and albuminuria. Conversely, rHA and eHA seem to exhibit improved diagnostic efficacy in the intermediate to advanced stages of the disease, enabling the better risk stratification of patients. This approach may allow the identification of those at a higher risk of disease progression in which a more aggressive pharmacological approach could be beneficial.

This study has both strengths and limitations. Among the former, an in-depth characterization of HA microheterogeneity was performed via MS analysis, which also allowed us to assess eHA and rHA levels to complement the more commonly clinically determined value of tHA in clinical settings. On the other hand, whereas HA structure determination can be performed in less than 15 min by LC-MS analysis, it must be noted that an idoneous LC-MS platform might not be available in all clinical settings for routine clinical analysis, although an increasing number of hospital laboratories have implemented analytical platforms and used them for routine analyses.

In conclusion, in this study, we demonstrated that rHA and eHA were significantly altered in DKD patients in a dose-dependent correlation with renal dysfunction and might be exploited to complement the diagnosis of kidney damage. eHA was identified as an independent predictor of renal impairment. The results prompt the need for further studies more deeply addressing biochemical processes leading to albumin changes and the clinical utility of eHA parameters for the diagnosis and prognosis of T2DM-related kidney disease.

## 4. Materials and Methods

### 4.1. Patients and Study Design

Patients were screened for study enrollment among those attending the outpatient clinic of the Metabolic Diseases & Clinical Dietetics Unit and the Nephrology, Dialysis and Transplantation Unit of the IRCCS Azienda Ospedaliero-Universitaria di Bologna (Italy). The inclusion criteria were age between 18 and 70 years, a diagnosis of T2DM for at least 1 year, and renal function at various stages (G1-5, with/without albuminuria), allowing coverage of DKD complications. Diabetes mellitus was diagnosed according to the American Diabetes Association criteria [41], i.e., fasting plasma glucose (FPG) ≥ 126 mg/dL (7.0 mmol/L) or 2 h plasma glucose ≥200 mg/dL (11.1 mmol/L) during a 75 g oral glucose tolerance test OGTT or glycated hemoglobin ≥6.5% (48 mmol/mol), or, in a patient with classic symptoms of hyperglycemia or hyperglycemic crisis, random plasma glucose ≥200 mg/dL (11.1 mmol/L).

Subjects were classified into four groups, NO-DKD (nonrenal impairment), DKD-I (moderately increased risk), DKD-II (high risk), and DKD-III (very high risk), on the basis of the severity of renal dysfunction using a combination of albuminuria levels, measured as the urine albumin–creatinine ratio (UACR), and the eGFR category, as recommended by KDIGO guidelines [30]. Demographic, anthropometric, and blood pressure parameters were assessed, and a medical history was collected to survey drug therapy. Blood samples (plasma and serum) were collected on fasting and were centrifuged at 3000× *g* for 10 min; the serum was aliquoted into cryotubes (Corning Inc., Corning BV, Amsterdam, The Netherlands) and stored at –80 °C until analysis. Renal function parameters were also recorded 1 year after study inclusion in all subjects. The study protocol was approved by the local ethical committee, and written informed consent was obtained from all patients before enrollment, according to the 1975 Declaration of Helsinki.

### 4.2. Bromocresol Green (BCG) Colorimetric Assay

The total albumin serum concentration (tHA) was determined by BCG colorimetric assay, adopting the well-established method currently used in clinic on a smaller scale [42]. The BCG reagent contains 0.2 mM BCG, 0.1 mM succinate buffer (pH 4.2), and 0.8% *v*/*v* Tween^®^20. Serum samples were diluted 5-fold in ultrapure water. A 5 µL aliquot of diluted serum sample was added to 200 µL of BCG reagent and gently mixed. The samples were incubated for 5 min at room temperature. Blank solutions were prepared in parallel and contained all of the components except for the plasma sample. Then, 200 µL of each sample and blank were transferred to a well of a clear 96-well flat-bottom microplate, and the absorbance in the range of 570–670 nm (at 620 nm) was measured using a Spark^®^ multimode microplate reader (Tecan, Grödig, Austria). HA quantitation was performed by interpolating the absorbance value at OD620 nm in a calibration curve built using HA standard solutions at increasing concentrations (5, 7, 5, 10, 15, and 20 mg/mL). A standard curve was generated with each set of assays. All assays were performed in triplicate.

### 4.3. Liquid Chromatography—Mass Spectrometry (LC-MS) Analysis

For the quantitation of albumin structural alterations, the validated high-resolution LC-MS method, previously reported by Naldi et al., was employed with minor modifications [43]. The method exploits high-performance liquid chromatography coupled to electrospray ionization/quadrupole time-of-flight mass spectrometry (HPLC-ESI-Q-ToF). Serum samples were diluted 1:100 with ultrapure water and filtered through a 0.22 μm syringe filter (Merck KGaA, Darmstadt, Germany). HPLC analyses were carried out on an Agilent 1200 HPLC System (Walbronn, Germany). The chromatographic separation of HA from other plasma proteins was achieved using a Phenomenex Jupiter C4 column (5 μm, 300 Å, 150 mm × 2.0 mm i.d.). A gradient was developed with mobile phases A [water/acetonitrile/formic acid (99/1/0.1, *v*/*v*/*v*)] and B [acetonitrile/water/formic acid (98/2/0.1, *v*/*v*/*v*)], as follows: 20–70% B, in 5 min; 70% B, for 1 min. In between injections, the column was equilibrated for 5 min with starting conditions. The flow rate was set at 0.4 mL/min, and the injection volume was 3 μL.

A quadrupole time-of-flight hybrid mass analyzer (Q-ToF Micro, Micromass, Manchester, UK) with a Z-spray electrospray ion source (ESI) was used for the MS analysis. The capillary voltage and cone voltage were set at 3.0 kV and 40 V, respectively. The ESI-Q-ToF source temperature was set to 150 °C, while the desolvation temperature was set to 300 °C. The scan time was set at 2.4 s, while the interscan time was set to 0.1 s. The desolvation gas flow was set at 1000 L/h, and the cone gas flow was 120 L/h. Total ion current (TIC) chromatograms were acquired in positive polarity in the range of 1000–1800 *m*/*z*. Using the maximum-entropy (MaxEnt1)-based software included with MassLynx 4.1 software, the HA baseline-subtracted spectrum (*m*/*z* 1084–1534) was deconvoluted into a genuine mass scale. The output parameters were as follows: mass range 61,500–71,500 Da and resolution 2 Da/channel. The relative abundances of HA forms were estimated from the intensity of each form (obtained from the deconvoluted spectrum) and are expressed as a percentage of the total intensity of all the forms. Microsoft Excel software version 2304 (Microsoft Corporation, Redmond, WA, USA) was used for the data analysis.

### 4.4. Assessment of eHA and rHA

eHA and rHA were calculated using the following formulae [21]:$$eHA\left(\frac{g}{dL}\right)=\frac{tHA\left(\frac{g}{dL}\right)\times nHA\left(\%\right)}{100}$$
$$rHA\left(\frac{g}{dL}\right)=\frac{tHA\left(\frac{g}{dL}\right)\times HMA\left(\%\right)}{100}$$
where nHA is the amount of native albumin, HMA is the amount of albumin forms with reduced Cys34 as assessed by LC–MS analysis, and tHA is the total albumin concentration as assessed by the BCG assay.

### 4.5. Statistical Analysis

Normally distributed data were reported as the mean and standard deviation (SD), whereas non-normally distributed parameters were summarized using the median and interquartile range. The distribution of the data was preliminarily assessed by the Kolmogorov—Smirnov test. Categorical variables were reported as the absolute frequency and percentage. When appropriate, comparisons between groups were tested by the unpaired Student’s t test or Mann—Whitney U test. For comparisons between three or more groups, Kruskal—Wallis analysis of variance (ANOVA) was performed, followed by a post hoc analysis in which Bonferroni’s correction for multiple comparisons was applied. The associations between clinical parameters and nHA, tHA, and eHA levels were evaluated by Spearman’s correlation analysis, while the sensitivity and specificity of the same parameters for diagnosing kidney damage were determined by receiver operating characteristic (ROC) curve analysis. The resulting AUCs were compared according to the DeLong method. Finally, a multivariable logistic regression analysis with backward selection was performed to compare the ability of rHA and tHA to predict renal damage. All tests were two-sided, and a *p*-value less than 0.05 was used to indicate statistical significance. The data were processed using GraphPad Prism 8.4.2 (GraphPad Software, San Diego, CA, USA) and the Statistical Package for the Social Sciences (SPSS version 25; IBM Corp., Armonk, NY, USA).

## Figures and Tables

**Figure 1 ijms-25-03168-f001:**
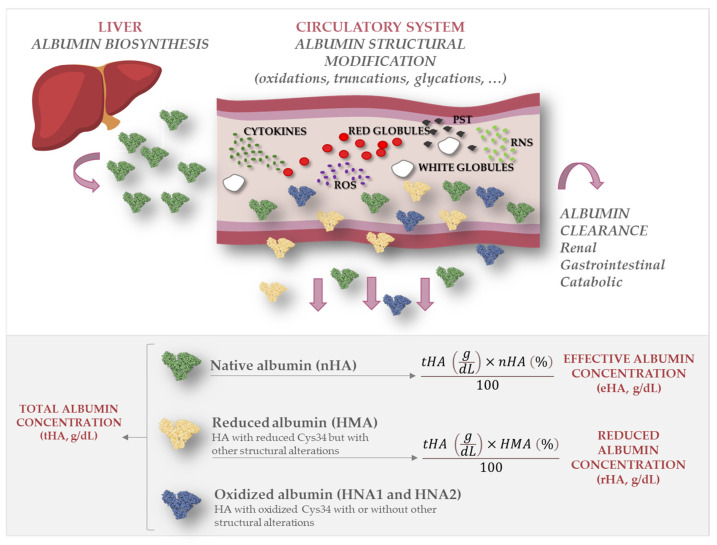
Total, reduced, and effective albumin. The figure depicts how native albumin (nHA) undergoes structural modifications such as oxidation, truncation, and glycation once it is secreted in the circulatory system, mainly due to oxidative stress and inflammatory processes. While total albumin concentration (tHA) refers to the serum concentration of all albumin forms, effective albumin (eHA) refers to the serum concentration of nHA and reduced albumin concentration (rHA) refers to the concentration of all forms with free Cys34.

**Figure 2 ijms-25-03168-f002:**
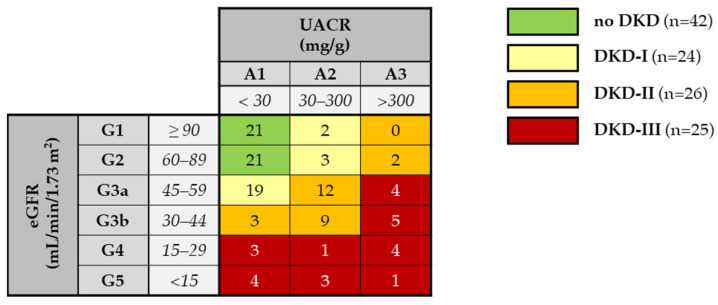
Patient classification according to the CGA scheme by KDIGO [30]. Green boxes: control, low risk (no CKD); yellow boxes: DKD-I group, moderately increased risk; orange: DKD-II group, high risk; red boxes: DKD-III group, very high risk. The number of enrolled patients in each eGFR category (G1–G5) and albuminuria category (A1–A3) is reported.

**Figure 3 ijms-25-03168-f003:**
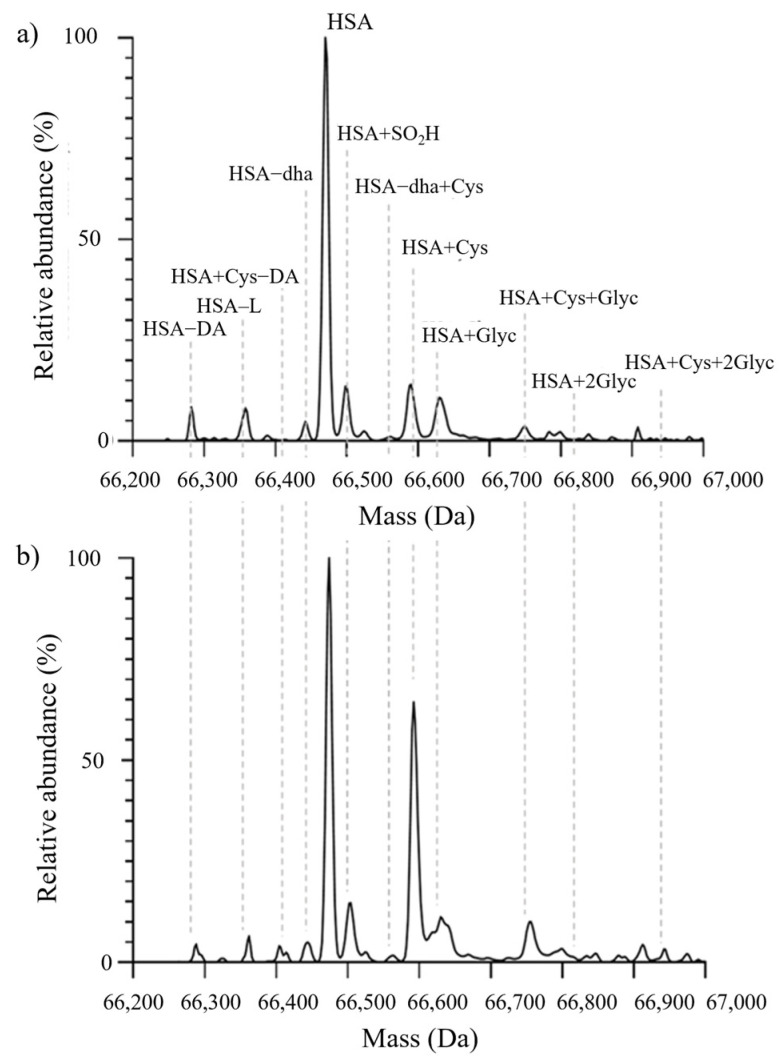
Representative MS spectra from a T2DM patient without renal impairment (**a**) and from a T2DM patient with renal impairment (**b**). Abbreviations: HA−DA: truncation at the N-terminal portion; HA−L: truncation at the C-terminal portion; HA+Cys−DA: N-terminal truncated form cysteinylated at Cys34; HA: native albumin; HA−SO_2_H: albumin sulfonylated at Cys34; HA+Cys: cysteinylation at the level of Cys34; HA+Glyc: mono-glycation; HA+Cys+Glyc: cysteinylated form carrying one glycation; HA+2Glyc: di-glycation; HA+Cys+2Glyc: cysteinylated form carrying two glycations.

**Figure 4 ijms-25-03168-f004:**
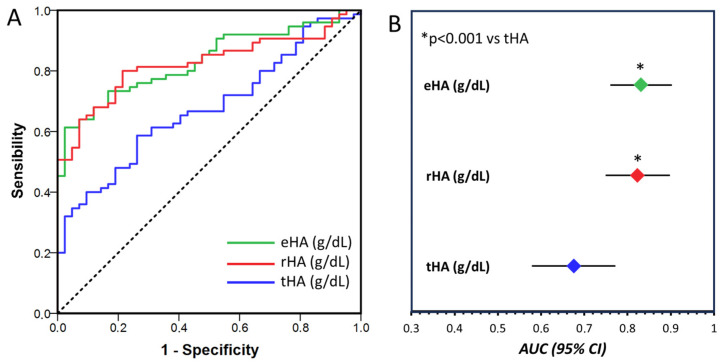
(**A**). Receiver operating characteristic curve analysis of total HA concentration (tHA, blue line), reduced HA concentration (rHA, red line), and native albumin concentration (eHA, green line) for the diagnosis of DKD. Dashed line represent the reference line. (**B**). Area under the curve (AUC) and 95% confidence interval for tHA (blue dot), rHA (red dot) and eHA (green dot). The AUCs were compared according to the DeLong method.

**Figure 5 ijms-25-03168-f005:**
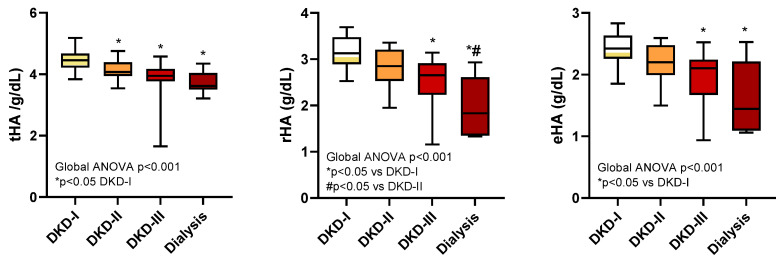
tHA, rHA, and eHA levels in diabetic patients with renal impairment (the DKD-I, DKD-II, DKD-III, and dialysis groups).

**Table 1 ijms-25-03168-t001:** Anthropometric and clinical data of T2DM patients with (DKD) and without (No DKD) renal impairment.

	No DKD *n* = 42	DKD *n* = 75	*p*-Value
**Anthropometric data**			
Age (years)	63 (56–67)	70 (64–74)	<0.001
Male sex (*n*, %)	29 (69)	52 (69)	1.000
BMI	28.7 (25.8–34.4)	32.3 (27.4–36.5)	0.114
**Drug therapy**			
Anti-hypertensives (*n*, %)	31 (74)	62 (83)	0.340
*ACE inhibitors (n, %)*	*15 (48)*	*29 (47)*	*0.883*
*Angiotensin receptor blockers (n, %)*	*13 (42)*	*27 (44)*	*0.882*
*Diuretics* (*n*, %)	*6 (14)*	*32 (43)*	*0.002*
Metformin (*n*, %)	29 (70)	42 (56)	0.175
Insulin (*n*, %)	15 (36)	40 (53)	0.083
Other glucose-lowering therapies (*n*, %)	22 (52)	43 (57)	0.699
*Sulfonylureas (n, %)*	*4 (18)*	*10 (23)*	*0.222*
*DPP-4 inhibitors (n, %)*	*7 (32)*	*13 (30)*	*0.896*
*GLP-1 Receptor agonists (n, %)*	*8 (36)*	*21 (49)*	*0.298*
*SGLT-2 Inhibitors (n, %)*	*5 (12)*	*3 (4)*	*0.067*
Statin/fibrates (*n*, %)	26 (62)	50 (67)	0.687
**Biochemical parameters**			
HbA1c (%)	7.0 (6.4–7.6)	7.1 (6.2–7.6)	0.952
Glucose (mg/dL)	133 (110–155)	123 (110–151)	0.496
Total cholesterol (mg/dL)	172 ± 35	149 ± 31	0.001
HDL (mg/dL)	47 ± 9	41 ± 9	0.005
LDL (mg/dL)	101 ± 31	79 ± 27	0.002
Triglycerides (mg/dL)	122 (87–177)	166 (116–204)	0.032
Triglycerides/HDL ratio	2.3 (1.5–3.7)	3.7 (2.8–5.5)	<0.001
Creatinine (mg/dL)	0.8 (0.8–0.9)	1.4 (1.2–1.7)	<0.001
UACR (mg/g)	7 (5–10)	66 (17–229)	<0.001
eGFR (mL/min/1.73 m^2^)	89 (77–97)	48 (34–56)	<0.001

BMI: body mass index; ACE: angiotensin-converting enzyme; DPP-4: dipeptidyl peptidase IV; GLP-1: glucagon-like peptide-1; SGLT-2: sodium-glucose cotransporter-2; HDL: high-density lipoprotein; LDL: low-density lipoprotein. UACR: urinary albumin-to-creatinine ratio. The data are reported as the mean and standard deviation, median and interquartile range, or absolute number and frequency.

**Table 2 ijms-25-03168-t002:** Relative abundances (%) of native and altered forms of human albumin (HA) in diabetes patients without renal damage (no DKD *n* = 42) and with renal damage (DKD, *n* = 75) determined by LC-ESI-MS analysis. The data are statistically expressed as medians and interquartile ranges.

HA Forms	Relative Abundance (%)		*p*-Value *
	No DKD *n* = 42	DKD *n* = 75	
nHA	59.0 (57.1–60.6)	52.5 (48.3–56.7)	<0.0001
HMA	75.7 (73.2–78.4)	67.1 (62.5–72.7)	<0.0001
HNA1	16.4 (13.7–19)	25.7 (19.8–30.9)	<0.0001
HNA2	8.9 (8.1–9.6)	7.9 (7–8.6)	<0.0001
Truncated	6.5 (5.1–7.9)	5.2 (4.4–7)	0.0242
Glycated	10.8 (9.2–12.2)	12.5 (10.5–14.7)	0.0028

* Mann—Whitney U test. Abbreviations: HMA: mercaptoalbumin; HNA1: non-mercaptoalbumin type 1; HNA2: non-mercaptoalbumin type 2.

**Table 3 ijms-25-03168-t003:** nHA, tHA, and eHA values for patients with or without diabetic kidney disease (DKD). Data were reported as medians and interquartile ranges. The *p*-value was determined and is reported.

HA Forms	Relative Abundance (%)		*p*-Value *
	No DKD *n* = 42	DKD (I–III) *n* = 75	
tHA (g/dL)	4.3 (4.1–4.5)	4.1 (3.9–4.4)	0.001
rHA (g/dL)	3.3 (3.2–3.4)	2.8 (2.5–3.1)	<0.001
eHA (g/dL)	2.5 (2.4–2.7)	2.2 (1.9–2.4)	<0.001

* Mann—Whitney U test. Abbreviations: tHA, total albumin; rHA: reduced albumin; eHA: native albumin.

**Table 4 ijms-25-03168-t004:** Correlations of tHA, rHA, and eHA levels, assessed at baseline, with creatinine, albuminuria, and eGFR (*n* = 117). Analysis was performed with the data collected both at baseline and at the one-year follow-up.

		tHA	rHA	eHA
		Rho (*p*-Value)	Rho (*p*-Value)	Rho (*p*-Value)
Baseline	Creatinine (mg/dL)	−0.260 (0.006)	−0.567 (<0.001)	−0.548 (<0.001)
	eGFR	0.337 (<0.001)	0.659 (<0.001)	0.649 (<0.001)
	Albuminuria	−0.497 (<0.001)	−0.546 (<0.001)	−0.554 (<0.001)
Follow-up	Creatinine (mg/dL)	−0.172 (0.077)	−0.508 (<0.001)	−0.509 (<0.001)
	eGFR	0.172 (0.076)	0.562 (<0.001)	0.584 (<0.001)
	Albuminuria	−0.363 (<0.001)	−0.461 (<0.001)	−0.407 (<0.001)

Abbreviations: tHA, total albumin; rHA: reduced albumin; eHA: native albumin.

## Data Availability

Data is contained within the article and Appendix A.

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
