# Peer review of "Association between Albumin Alterations and Renal Function in Patients with Type 2 Diabetes Mellitus"

_ijms, 2024, doi:10.3390/ijms25063168_

Round 1

Reviewer 1 Report

Comments and Suggestions for Authors

In the Nugnes et al. study, the authors evaluated the concentrations of effective albumin (HA) and reduced HA in a cohort of Type 2 Diabetes Mellitus (T2DM) patients with and without proteinuria. They correlated their findings with biochemical parameters and kidney impairment. The study is intriguing and contributes significantly to knowledge in the field. Please address the following comments before proceeding.

 Major Comments:

1)      In the Introduction section, page 2, lines 90-93, the study's aim should be stated objectively, and the use of parentheses should be minimized. Additionally, provide an explanation for what is meant by the "early" diagnosis of renal impairment. Clarify if this is related to the DKD phenotype.

 2)      Without a kidney biopsy, we cannot definitively assert that the "No DKD" group does not exhibit signs of DKD. Oshima M et al. (Nature Reviews Nephrology, 2021) identified four clinical presentations of DKD, including proteinuric DKD and no proteinuric DKD (UACR < 30 mg/g). Adjust the nomenclature and analyses accordingly. Include demographic data, such as the duration of diabetes mellitus (DM), frequency of microvascular-related DM complications (retinopathy and neuropathy), and macrovascular-related DM complications (stroke, TIA, IM, among others), to reinforce the diagnosis of DKD. Additionally, note that almost 40% of T2DM individuals may present with DKD without proteinuria. Revise the manuscript accordingly.

 3)      Specify how patients were initially diagnosed with DM, indicating whether glycated hemoglobin and/or fasting blood glucose were used.

 4)      Consider assessing the HDL/TG ratio, a marker of insulin resistance and cardiovascular risk, to provide insightful information about the groups in relation to albumin concentration, biochemical parameters, and kidney impairment.

 5)      Include a paragraph outlining the potential limitations of the study.

Minor Comments:

1)      Review the grammar of the sentence: "In addition, the levels of both eHA, representing the absolute concentration of nHA, and rHA, representing the absolute concentration of all albumin forms with reduced Cys34, were significantly lower in T2DM patients with kidney damage and were able to discriminate those patients from the control group better than tHA…" (lines 219-221, page 6).

2)      Describe the abbreviations HMA (mercaptoalbumin), HNA1 (non-mercaptoalbumin type 1), HNA2 (non-mercaptoalbumin type 2) when they first appear in the manuscript, along with rHA.

Author Response

Please see the file attached.

Reviewer 2 Report

Comments and Suggestions for Authors

Dear authors, The article is well presented.

1. The main question of the work is devoted to the search for a highly accurate criterion for assessing the progression of chronic kidney disease. The authors propose to build on the known criterion of albuminuria, but at the same time highlight its different fractions, which probably carry different informative significance.
2. Based on the existing literature analysis, highlighting the prognostic significance of albumin fractions to evaluate the prognosis of chronic kidney disease is original data.
3. The novelty of the present study is an attempt to highlight the informative significance of all three albumin fractions and to link them with the course of chronic kidney disease and its progression. Of course, this is very interesting material
4. The research certainly requires further continuation. From a scientific point of view, the authors managed to find certain patterns and show their significance and reliable differences. On the other hand, the clinical significance of the study is not yet sufficient for recommendations for practice. Criterion values of the most informative indicator are needed; a simple implementation for practice is needed
5. The conclusions are consistent with this study. Without stopping at the results obtained, the authors raise further substantiated questions that await their implementation and, above all, with an eye to clinical nephrological practice.
6. The literature corresponds to the presented materials. As usual, I would like to expand it, but the authors stopped at a sufficient number of sources
7. Tables and drawings in sufficient quantity. Considering the complexity of the topic for a scientific audience, I would recommend adding a visualization of the formation of various types of albuminuria, which are given in the article, namely: how they are formed, how they are metabolized and why these albumin fractions were chosen by the authors.

As a clinician, I would like to understand the relationship between each other in metabolism, origin, degradation of tHA, rHA and eHA, maybe present them in the form of a picture? And how can I use your very interesting experience in clinical practice?

Author Response

Please see the file attached.

Reviewer 3 Report

Comments and Suggestions for Authors

First of all, we would like to thank the authors for the preparation of a manuscript, which evaluates  the Association between albumin alterations and renal function in patients with T2DM

A practical, well-documented paper, with a correct methodology and a more than relevant discussion, suggest the importance of considering not only the amount of circulating albumin, but also its structural integrity, and recommends the need to investigate the diagnostic capacity of these parameters.

However, I would like to make some recommendations to improve the quality of the manuscript.

*In the ¨Introduction¨ section:

1.- Lines 50-51: review well the definition of KDIGO ¨decrease in renal function with an estimated glomerular filtration rate (eGFR)< 60 mL/min/1,73 m2 or the presence of markers of renal impairment, such as albuminuria, hematuria or abnormalities detected by laboratory or imaging tests, present for at least for at least 3 months and with health consequences¨

I recommend reviewing this paper:

García-Maset R, Bovera J, Segura de la Morena J, Goicoechea M, Cebollada del Hoyo BJ, Escalada J, et al. Documento de información y consenso para la detección y manejo de la enfermedad renal crónica. Nefrologia. 2022;42(3):233–264

2.- Lines 56-57: ¨ Diagnostic renal biopsy is only required  when non-diabetic renal disease is suspected¨ this comment to modify it there are specific recommendations for the performance of renal biopsy in the patient with DKD.

I recommend reviewing these papers:

Bermejo S, González E, López-Revuelta K, Ibernon M, López D, Martín-Gómez A, et al. Risk factors for non-diabetic renal disease in diabetic patients. Clin Kidney J. 2020 Jan 3;13(3):380-8.

Bermejo S, García-Carro C, Soler MJ. Diabetes and renal disease-should we biopsy? Nephrol Dial Transplant. 2019:gfz248

3.- Lines 85-86:  ¨¨hypoalbuminemia is considered a reliable clinical indicator of DKD and is associated with impaired renal function and poor prognosis in T2DM patients with renal damage¨, this comment is not clear, hypoalbuminemia is a risk factor for end-stage CKD because it increases morbidity and mortality in renal failure. Please modify it.

4.- Line 90: introducing the definition of rHA in the previous paragraph, it is not clear to put it in this paragraph, it tends to err with the other types of HA and confuses the reading of the manuscript.

*In the ¨Results¨¨ section:

1.- Lines 100-102: update bibliography to KDIGO 2022 guidelines

KDIGO Diabetes Work Group. KDIGO March 2022 Clinical Practice Guideline for Diabetes Management in Chronic Kidney Disease. Kidney Int. 2022 Nov;102(5S):S1-S127. doi: 10.1016/j.kint.2022.06.008.

2.- Although Figure 1 is clear to be able to group the study patients into the 4 groups assigned according to the KDIGO classifications, however many problems arise when we want to classify a patient with DKD who has a GFR > 60 ml/min, ACR less than 30 mg/g and if the patients present any alterations in the sediment such as microhematuria or alterations in the imaging study. I recommend commenting on this.

3.- As the methodology section is at the end of the manuscript, and it is there where the acronyms of the albumin types (HNA1, HMA….) are described, please place it before mentioning it, to make the reading faster.

*In the ¨Materials and Methods¨ section:

Well written, clear and concise, no recommendations in this section.

*In the ¨Discusion¨ section:

Well written, clear and concise, no recommendations in this section.

Comments on the Quality of English Language

 Minor editing of English language required

Round 2

Reviewer 1 Report

Comments and Suggestions for Authors

The authors have partially addressed the question about demographic data between the two groups, such as the duration of diabetes mellitus (DM), frequency of microvascular-related DM complications (retinopathy and neuropathy), and macrovascular-related DM complications (stroke, TIA, MI, among others), to reinforce the diagnosis of DKD. Therefore, based on these findings, the authors may describe the patients as proteinuric DKD and non-proteinuric DKD, as a kidney biopsy is not available. Revise the manuscript accordingly.  

Author Response

We would like to really thank the reviewer for his/her suggestion, and we agree with him/her that the addition of further demographic information would greatly implement the diagnosis of DKD. However, patient enrollment was performed by clinicians (endocrinologists and nephrologists) and the requested demographic parameters were not directly included in the list of extracted demographic data at the time of the study, hence, retrieving them would require a quite extensive time which exceeds the given time for revision and possibly the general outcomes of the study. However, as requested, we better underlined in the result section 2.1 the information requested by the reviewer about the duration of the diabetes which was longer than a year for all patients enrolled in the study.